# Raman Spectroscopy and Thermoelectric Characterization of Composite Thin Films of Cu_2_ZnSnS_4_ Nanocrystals Embedded in a Conductive Polymer PEDOT:PSS

**DOI:** 10.3390/nano13010041

**Published:** 2022-12-22

**Authors:** Yevhenii Havryliuk, Volodymyr Dzhagan, Anatolii Karnaukhov, Oleksandr Selyshchev, Julia Hann, Dietrich R. T. Zahn

**Affiliations:** 1Semiconductor Physics, Chemnitz University of Technology, 09107 Chemnitz, Germany; 2Center for Materials, Architectures, and Integration of Nanomembranes (MAIN), 09126 Chemnitz, Germany; 3V.E. Lashkaryov Institute of Semiconductor Physics NAS of Ukraine, 03028 Kyiv, Ukraine; 4Physics Department, Taras Shevchenko National University of Kyiv, 60 Volodymyrs’ka str., 01601 Kyiv, Ukraine

**Keywords:** Cu_2_ZnSnS_4_ (CZTS), nanocrystals, PEDOT:PSS, thermoelectric properties

## Abstract

Cu_2_ZnSnS_4_ (CZTS) is an intensively studied potential solar cell absorber and a promising thermoelectric (TE) material. In the form of colloidal nanocrystals (NCs), it is very convenient to form thin films on various substrates. Here, we investigate composites of CZTS NCs with PEDOT:PSS, a widely used photovoltaics polymer. We focus on the investigation of the structural stability of both NCs and polymers in composite thin films with different NC-to-polymer ratios. We studied both pristine films and those subjected to flash lamp annealing (FLA) or laser irradiation with various power densities. Raman spectroscopy was used as the main characterization technique because the vibrational modes of CZTS NCs and the polymer can be acquired in one spectrum and thus allow the properties of both parts of the composite to be monitored simultaneously. We found that CZTS NCs and PEDOT:PSS mutually influence each other in the composite. The thermoelectric properties of PEDOT:PSS/CZTS composite films were found to be higher compared to the films consisting of bare materials, and they can be further improved by adding DMSO. However, the presence of NCs in the polymer deteriorates its structural stability when subjected to FLA or laser treatment.

## 1. Introduction

The recent trends in photovoltaic science and technology include searching for new materials for absorbing layers to replace silicon which is currently dominating the market [1,2,3]. Such materials should be cheap, environmentally friendly and have a direct band gap with a value that ensures good absorption in the solar radiation range. Silicon is non-toxic and widely available material, but its production for photovoltaic purposes is very energy consuming. Moreover, due to its indirect bandgap, large film thicknesses are needed to ensure complete absorption of solar radiation, precluding transition to a “true” thin-film technology or using flexible substrates. Among the alternative absorber materials which meet all of the above requirements are Cu_2_ZnSn(S,Se)_4_ (CZTSSe) and related compounds [2,4,5,6,7,8]. The possibility of synthesizing such materials by colloidal synthesis makes them attractive for thin-film technologies [2,4,9,10], and their alternative potential applications include electrodes for perovskite solar cells [11], humidity sensors [12], photocatalysis [13], as well as thermoelectrics [14,15]. However, one of the main obstacles to the application of most nanocrystal (NC) materials is the poor quality of the NC films and the resulting low electrical conductivity. A solution to this problem may lie in forming composites of NCs with polymers. In this work, we investigate, for the first time, the composites of CZTS NCs with PEDOT:PSS, a well-known p-type conductive polymer, which has found wide application not only as a transparent electrode in electronics [16,17,18] and photovoltaics [17,19,20] but also as an active layer in thermoelectrics [21,22,23]. Taking into account that CZTS is a good solar light absorber, PEDOT:PSS is conductive and a good thermoelectric (TE) material, the combination of these materials can be promising, not only for photovoltaic and TE applications separately but also in their combination, i.e., towards solving the existing challenge of converting the heat generated by solar cells into electricity as well.

In this work, we present the results of a Raman spectroscopy study of composite films prepared from an aqueous colloidal solution of CZTS NCs mixed in different proportions with a PEDOT:PSS solution and deposited onto a glass substrate. Raman spectroscopy was chosen here as the main characterisation method because it has proven to be very convenient and informative for CZTS-like compounds, and their phonon spectra deliver structural information not only about the main phase itself but also about possible secondary phases [24,25,26,27,28,29,30,31,32]. Furthermore, the range of the characteristic modes of PEDOT:PSS can be registered in the same spectrum. The parameters of the film deposition were optimized to avoid aggregation of the NCs and the formation of secondary phases in them [26]. We also investigated the effect of flash lamp annealing (FLA, also known as intense pulsed light, IPL, annealing) on the structural, electrical, and TE properties of the films. For thin films, this technique is a promising alternative to thermal annealing [33,34]. To the best of the authors’ knowledge, there are no studies as yet on the FLA treatment of PEDOT:PSS and CZTS NCs in the PEDOT:PSS matrix. In addition, we performed a preliminary study of the thermoelectric properties of the CZTS NCs films and their composite with PEDOT:PSS, which have also not yet been reported.

## 2. Materials and Methods

The synthesis of CZTS NCs was carried out following the known procedure described by us previously [24,35]. After the synthesis, the solution of CZTS NCs obtained was purified from residual reaction products by the addition of isopropanol and centrifugation. Afterwards, the precipitated CZTS NCs were redissolved in deionized water.

PEDOT:PSS was purchased from Heraeus (Clevios PH 1000) (Heraeus, Leverkusen, Germany).

Flash lamp annealing was carried out using the FLA setup from Dresden Thin Film (DTF) Technology GmbH (SOLAYER GmbH, Kesselsdorf, Germany) based on a Xenon lamp emitting in the range of 300 to 800 nm in a glove box system with a nitrogen atmosphere. 

Raman spectra were excited using a 514.7 nm solid-state laser (Cobolt) and were registered at a spectral resolution of about 2 cm^−1^ using a LabRam HR800 equipped with cooled CCD detectors (HORIBA Europe GmbH, Dresden, Germany). The incident laser power under the microscope objective (100×) was in the range of 0.01–0.1 mW.

Thermoelectric and electric characterizations were performed using an “IPM-SR7 Power factor measurement system” customized for films and bulk materials in the temperature range of 300 to 900 K (Fraunhofer IPM, Freiburg, Germany). For the thermoelectric measurement, the samples were prepared using spin-coating deposition onto glass substrates. Glass was chosen as a substrate due to its low thermal and electrical conductivity, minimizing the influence of the substrate on measurements. To correctly determine the electrical conductivity, the thickness of each sample was measured using atomic force microscopy (AFM Agilent 5500) (Agilent Technologies, Inc., Santa Clara, CA, USA). To do this, a scratch was made on the film using a sharp object and a force enough to reach the substrate without damaging it. The AFM profile on the scratch is then measured and the depth (height) of the scratch is considered as the thickness of the film.

## 3. Results and Discussion

### 3.1. Preparing Composites of CZTS NCs with PEDOT:PSS and Their Raman Spectra

Concentrated aqueous solutions of CZTS NCs were mixed with PEDOT:PSS with a CZTS NC content (volume ratio) ranging from 0% (pure PEDOT:PSS) to 100% (only NCs) in steps of 10%. The obtained solutions were mixed by shaking and deposited by drop-casting onto a glass substrate. The preparation of the mixtures should be performed just before the film deposition because the “gelling” of the mixtures occurs less than an hour after preparation. The gelation occurred even faster if the mixture was ultrasonicated. The completed “gelling” process resulted in a clear separation into a transparent liquid and a single piece of black jelly. The latter did not dissolve again even with intensive shaking and also kept its shape even after being taken out of the solution for several hours, but after long drying under a hood, it almost completely dried out (Appendix A).

The transparent liquid did not give any signals in the Raman spectra, even after drying a drop of it on the substrate. The spectrum of the gel differs from the spectrum of the fresh mixture, namely the intensities of the PEDOT peaks are much lower, and PSS peaks are barely visible at all (Appendix A). Based on the physical properties of the gel as well as Raman spectroscopy data, it can be hypothesized that the NCs interact with PEDOT, resulting in a detachment of a part of the PSS molecules, which probably remain in the supernatant after the formation of the black gel (presumably consisting of PEDOT and CZTS NCs). Alternatively, it can be assumed that the NCs captured in the matrix of PEDOT:PSS, together with some amount of water, resulted in a jelly-like structure. Establishing the exact mechanism needs further investigations which are beyond the scope of this paper. Interestingly, if the solution is applied before the gelling process begins, the resulting film remains stable for months. The film deposition and drying conditions were the same as established by us in our previous work [26], being optimal for the formation of CZTS NC films without a noticeable content of secondary phases. The inspection of the films under an optical microscope revealed a network of cracks for pure CZTS NC films (Figure 1a), whereas the addition of a small amount (e.g., 10%) of PEDOT:PSS allows the formation of continuous films (Figure 1b).

Figure 2a shows the Raman spectra for CZTS NCs/PEDOT:PSS blends of different ratios. The Raman spectrum of pure CZTS NCs reveals a characteristic peak at 332 cm^−1^, and no features related to secondary phases are observed [24,26,36]. The good crystallinity of the synthesized NCs is confirmed by observing distinct overtones of the main CZTS peak at about 650 and 1000 cm^−1^. The spectrum of pure PEDOT:PSS exhibits a set of characteristic peaks in good agreement with data in the literature [19,21,23,37,38,39,40]. The difference in the position of the characteristic bands of PEDOT:PSS are not only caused by changes in the structure but also depend on the excitation wavelength [41]. The features of the Raman spectrum can provide information about oxidation, doping, and the organisation of the PEDOT:PSS [37].

As can be seen in Figure 2, when CZTS NCs are added to PEDOT:PSS, the changes in the Raman spectra start to become visible at 20% CZTS NCs with a slight intensity increase around 328 cm^−1^ and are clearly visible at 30% CZTS NCs with a distinct CZTS peak (Figure 2b). Its position is, however, shifted by 4 cm^−1^ downwards from the 332 cm^−1^ position for pure CZTS NC film. With further increases in the concentration of CZTS NCs, a gradual shift of the characteristic band of CZTS NCs to higher frequencies is observed, and at a concentration of 90%, it is located at 332 cm^−1^ as in the pure CZTS NC film. In addition, the presence of a second (~650 cm^−1^) and third overtone (~1000 cm^−1^) in the spectra can be clearly observed, indicating that the CZTS NCs retain their good crystallinity after embedding into the polymer matrix.

Two possible reasons for the CZTS Raman peak position downward shift in the composite can be considered: (i) an interaction between CZTS NCs and the PEDOT:PSS polymer matrix, or (ii) a spectral overlap of the 318 cm^−1^ band of PEDOT:PSS and the 332 cm^−1^ band of CZTS NCs, resulting in an apparent single peak with an intermediate position, i.e., at 328 cm^−1^. In the latter case, however, one would expect an increase in the half-width of the resulting peak with respect to that of the CZTS mode or the polymer mode. Moreover, no shift at the high concentration of CZTS NCs would occur in case (ii), since the contribution of the low-intensity 318 cm^−1^ band would only appear as a shoulder of the CZTS peak. Therefore, factor (i) seems to be more probable in our case, although both (i) and (ii) may contribute, in general, so the shift of the CZTS Raman peak in the composite with PEDOT:PSS requires further detailed investigations in the future. In particular, it would be helpful to investigate NCs with different compositions (e.g., alloying CZTS with Se or Te, or partial cation substitution) in order to preclude their Raman peak from overlapping with the 318 cm^−1^ peak of PEDOT:PSS (e.g., the main Raman peak of selenide is expected at 184 cm^−1^ [25] and of the telluride, at 160 cm^−1^ [10]). 

The vibrational features of the polymer were also found to be affected in the composite with CZTS NCs (Figure 2b). The strongest band of PEDOT:PSS was observed at 1438 cm^−1^ and assigned to C_α_ = C_β_ bond vibrations [19,37,39,40]. This band shifted down to 1434 cm^−1^ in the sample with 90% of CZTS NCs. A much more pronounced shift was observed for another mode that shifts from 1495 upwards to 1515 cm^−1^ (Figure 2b). Furthermore, the mode observed at 1633 cm^−1^ in the pure polymer and the sample with 10–30% of NCs disappears at higher NC contents. All these changes in the spectrum of the polymer can be related either to changes in the structure or conformation of the polymer or to NC–polymer interaction. As both factors can mutually influence each other, distinguishing between them is hardly possible within the scope of the present study. Nevertheless, we can still make some assumptions about the structural transformations in the polymer or change in the charge distribution over its parts, based on the previous Raman reports in the literature, in particular, with respect to a possible change in the structure of PEDOT:PSS from benzoid to quinoid [19,21,39,40], or transition from a bipolaron to a polaron and a neutral state as a result of a changed doping level due to electronic interaction with the NCs [41]. It should also be kept in mind that CZTS is a highly photoactive material, with a high absorption coefficient at the wavelength of the Raman excitation (514.7 nm). Therefore, a photostimulated generation of free charge carriers in the NCs can cause in situ (transient) changes in the structure of the polymer, which do not take place in the composite without laser illumination.

The NC solutions used for the preparation of composites with PEDOT:PSS were subject to a standard purification procedure that removed most of the residuals of the synthesis, which are dissolved in water, such as NaCl and other salts, and an excess of thioglycolic acid (TGA) ligand molecules (i.e., ligand molecules not bound to the NC surface). Nevertheless, a certain amount of free ligand molecules can still be present in the solution (as a result of their detachment from the NC surface) and may have their own effect on the PEDOT:PSS. In order to check this, a control sample was prepared as a TGA solution in water, at a concentration that can be expected in the samples of CZTS NCs. Figure 3 shows the effect of the pure solution of TGA diluted to different concentrations on the Raman spectrum of PEDOT:PSS. A downward shift of the main peak very similar in magnitude was observed in this case, indicating that, in the mixtures with TGA-stabilized CZTS NCs, the downward shift was most likely caused by the TGA molecules as well. The range of 1450–1650 cm^−1^ is, on the contrary, not affected by the TGA solution. Therefore, the changes observed in this spectral range of PEDOT:PSS mixed with CZTS NCs are most likely induced by the NCs (Figure 2). The effect of concentrated TGA on the Raman spectrum of PEDOT:PSS was found to be completely different (Appendix A) from the spectral changes observed for the composites (Figure 2) or diluted TGA solution (Figure 3), and revealed tremendous changes in the polymer structure, which do not take place in the composite samples.

Control samples were also prepared by mixing PEDOT:PSS with NaOH to check the influence of different pH values, and with supernatant (liquid remaining after purification of the NCs by centrifugation) to check a possible influence of residual salts. In all these cases, the main characteristic PEDOT:PSS band exhibited qualitatively the same shift as for adding TGA, but the changes in other spectral regions were slightly different for different control samples. Nevertheless, the main conclusion that can be drawn from studying the control samples is that the spectral effects in the Raman spectra of PEDOT:PSS/CZTS NCs composites can at least partially be attributed to the interaction between the polymer and NCs. Note that the authors of [42] did not observe an influence of NaOH on the spectrum of PEDOT:PSS, but this may be due to some differences in the concentrations or other experimental conditions (including the measurement conditions of the Raman spectra).

### 3.2. Flash Lamp Annealing of Composite Films

We found earlier [36], that for CZTS NCs obtained by the same synthesis as in this work, the FLA treatment with energy densities in the range of 8–15 J/cm^2^ is optimal for improving the crystallinity of the CZTS phase without the formation of a significant volume of secondary phases. A power density below 8 J/cm^2^ did not lead to a noticeable improvement in the crystallinity, while above 15 J/cm^2^ the formation of secondary phases was observed. Therefore, in this work, an energy density of 12 J/cm^2^ was chosen for the FLA treatment of PEDOT:PSS/CZTS NC composites. Figure 4a shows the Raman spectra of annealed pure CZTS NCs and PEDOT:PSS films, studied as reference samples for establishing the possible mutual influence of both components in the composites.

The spectrum of FLA-treated CZTS NCs (Figure 4a) exhibited an upward shift of the main band (334 cm^−1^) compared to the pristine material (331 cm^−1^) indicating an improvement in the crystallinity of the compound, an expected consequence of FLA treatment. Moreover, no additional spectral features appeared, which could be attributed to secondary phases formed during FLA. This result was in agreement with our previous work and confirms the choice of an energy density of 12 J/cm^2^ as optimal. The appearance of an increasing background in the high-frequency region of the spectrum probably stemmed from the photoluminescence of amorphous carbon species [43] formed due to the partial decomposition of the ligands. The weak broad feature that appeared around 1600 cm^−1^ due to Raman scattering of such amorphous carbon supports this assumption [44]. The spectrum of pure PEDOT:PSS almost remained unaffected by FLA indicating the high stability of PEDOT:PSS at this FLA energy density. The film morphology also remained unchanged, as assessed by optical microscopy (Appendix A). For the PEDOT:PSS film, even with small incorporated amounts (e.g., 10%) of CZTS NCs, two types of spectra are observed, as presented in Figure 4b. In most of the areas of the film a broad feature typical for amorphous carbon was observed (point 1), while the vibrational spectrum of PEDOT:PSS was preserved only in some points (point 2). With an increasing NC content in the film, the dominance of the amorphous carbon spectrum across the film area increased, while the spectra of the polymer, observed in some spots (see spectrum for 70% NCs in Figure 4b), become weaker than that at low NC contents.

At the same time, the main phonon peak of CZTS NCs increased in intensity with increasing NC content, which was expected, while its spectral position reached a value of 335 cm^−1^ (for 90% NC content), which was higher than that of pure CZTS NC film (334 cm^−1^, Figure 4a), and the peak width was smaller (33 cm^−1^) than in the pure NC film (51 cm^−1^). These facts indicate a higher crystallinity of the NCs irradiated in the polymer. The heat transferred from the NCs to the polymer, as well as created by light absorption in the polymer itself, may be responsible for its amorphization, as revealed by the Raman spectra. An additional contribution to the amorphization of the polymer could be due to the deterioration of its thermal conductivity because of the incorporation of the NCs.

In such a way, the assumption can be made that the presence of CZTS NCs in PEDOT:PSS strongly affects the behaviour of the polymer during the FLA treatment. The difference in the Raman spectra measured in different regions could have been caused by an inhomogeneous distribution of CZTS NCs in the composite.

### 3.3. The Effect of Laser Power

Additional confirmation of our above assumptions made concerning FLA treatment was obtained from in situ laser irradiation experiments. For this purpose, the light intensity of the laser used for acquiring the Raman spectra was first reduced with neutral filters to a minimal value, at which it was still possible to obtain Raman features for both polymer and NCs. Afterwards, the intensity of the laser light was increased stepwise and the Raman spectra were recorded. As can be seen in Figure 5a–c, in the CZTS NCs/PEDOT:PSS composite, the decomposition of PEDOT:PSS started at a slightly lower intensity, and for CZTS NCs, at a slightly higher intensity compared to that of the pure materials. Thus, the results obtained with FLA and laser treatment correlated with each other.

Even though the observed destructive effect of the NCs on the PEDOT:PSS may appear negative at first glance, it may indicate that the presence of the NCs reduces the thermal conductivity in the PEDOT:PSS film. This effect, when made controllable, could be advantageous for TE applications. Higher electrical conductivity, on the contrary, is another advantage for a TE material. It is known that one of the efficient ways to improve the electrical properties of PEDOT:PSS is adding so-called “secondary doping” agents such as dimethyl sulfoxide (DMSO) [21,39,40,45]. For this reason, we prepared a series of composite samples of CZTS NCs mixed with PEDOT:PSS + DMSO, where the concentration of DMSO was 5 vol%, which was established as the optimal concentration from our previous studies [20]. Such a small amount of DMSO did not lead to any changes in the Raman spectra (not shown), indicating the absence of any noticeable effect on the structure of the NCs and the polymer. At the same time, the TE properties of the composite films prepared from NC/polymer solutions containing DMSO improved (Table 1). Therefore, we observed changes in the electrical properties but did not observe any changes in the Raman spectra after DMSO addition. Upon NC addition, however, we observed changes in both Raman spectra and electrical properties. A probable explanation for this is that NCs influence the electronic structure of the polymer while DMSO just leads to an ordering of the PEDOT:PSS molecules.

### 3.4. Thermoelectrical Studies

As mentioned earlier, both PEDOT:PSS and CZTS NCs are perspective materials not only for photovoltaic but also for thermoelectric applications. In terms of future applications in renewable energy harvesting, if the material has both good photovoltaic and thermoelectric properties, the possibility of using the same layer in the final device to generate electricity from both effects will make the panels thinner than if additional materials and layers are used. For this purpose, we investigated the TE properties of the composite CZTS NCs/PEDOT:PSS film in comparison with those of pure polymer and NC films. The samples were prepared using a spin-coating technique to obtain a smooth homogeneous film onto a glass to avoid any shortcuts or current flows through the substrate. Then, the conductivity, Seebeck coefficient, and resulting power factor were measured. Some of the preliminary results for the TE properties of PEDOT:PSS, CZTS NCs, and their composite are shown in Table 1.

One can see that the addition of NCs improves the TE properties of the PEDOT:PSS films. It appears that the addition of some amount of CZTS NCs to the PEDOT:PSS leads to the improvement of both conductivity and the Seebeck coefficient. Such an improvement, together with Raman study results, can be additional evidence of interaction between CZTS NCs and PEDOT:PSS in forming a composite material, but not just a mixture of independent CZTS NCs and PEDOT:PSS chains. The superior TE properties of the composite over those of the bare NCs and polymer are more pronounced for the corresponding FLA-treated films. The conductivity of the films increases in orders of magnitude in the FLA-treated films compared to pristine ones. This, along with the increasing Seebeck coefficient, leads to a significant enhancement of the power factor. An additional improvement of the TE properties of the composite is reached when DMSO was added to the blend. Therefore, the system seems to be promising for further investigations and the optimization of its TE properties by the systematic variation of the ratio of the components (PEDOT:PSS, CZTS NCs, DMSO), film thickness, and method of film fabrication, e.g., spin or spray coating.

## 4. Conclusions

The Raman spectra for CZTS NCs in the PEDOT:PSS matrix with CZTS NCs concentration from 0% to 100% were obtained and analysed. While cracks were observed in the film of pure CZTS NCs, even a small (10%) addition of PEDOT:PSS to the CZTS NC solution led to the formation of a continuous film, which is important for device applications for CZTS NCs. With a varying concentration of CZTS NCs in the PEDOT:PSS matrix, changes occur in both materials, as evidenced by the changes in their Raman peaks compared to the pure components.

The FLA treatment of CZTS NCs in PEDOT:PSS with a concentration of CZTS NCs ranging from 0% to 100% was performed. At a power density of 12 J/cm2, which was previously established as optimal for improving the crystallinity of CZTS NC films, the PEDOT:PSS was found to be stable. However, even a small content of CZTS NCs (10%) caused degradation of the PEDOT:PSS under the same FLA conditions, as revealed by the appearance of characteristic amorphous carbon bands in the Raman spectra. The NCs, on the contrary, achieved higher crystallinity upon FLA treatment of the composite, compared to the pure NC film. The data obtained so far allowed us to assume that CZTS NCs embedded into polymer have poorer heat dissipation and are thus strongly heated compared to pure NC film. Amorphization of the polymer in the composite may also be due to the heat transferred from the NCs. The CZTS NCs/PEDOT:PSS composite films show enhanced thermoelectrical properties compared to the pure PEDOT:PSS. A further improvement of the thermoelectrical properties can be achieved by FLA treatment and by the addition of DMSO.

## Figures and Tables

**Figure 1 nanomaterials-13-00041-f001:**
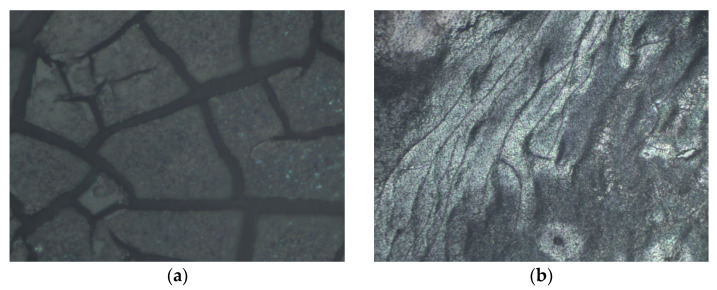
Optical images of (**a**) a CZTS NCs film, and (**b**) CZTS NCs with 10% of PEDOT:PSS added.

**Figure 2 nanomaterials-13-00041-f002:**
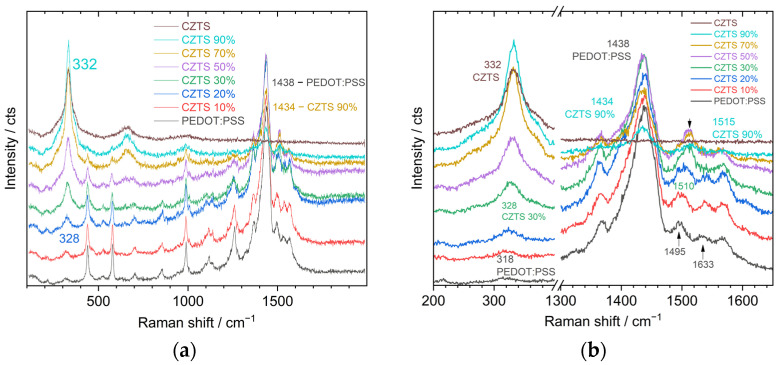
Raman spectra for (**a**) composite films with different contents of CZTS NCs in PEDOT:PSS: in the full spectral range of NCs and polymer vibrations, and (**b**) in the range of the main CZTS phonon and polymer bands.

**Figure 3 nanomaterials-13-00041-f003:**
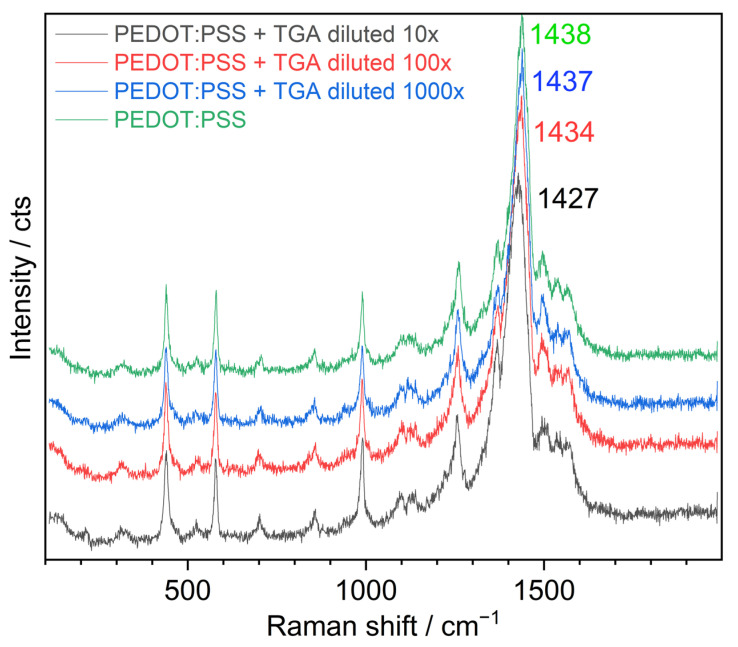
Control experiment showing the effect of pure TGA solution on the Raman spectra of PEDOT:PSS.

**Figure 4 nanomaterials-13-00041-f004:**
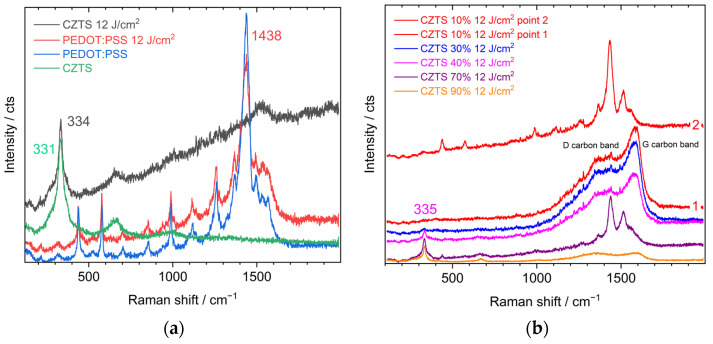
(**a**) Raman spectra for pristine CZTS NC and PEDOT:PSS films and the same films after FLA at 12 J/cm^2^; and (**b**) Raman spectra for the composite films with different ratios of CZTS NCs and PEDOT:PSS, subject to FLA at 12 J/cm^2^.

**Figure 5 nanomaterials-13-00041-f005:**
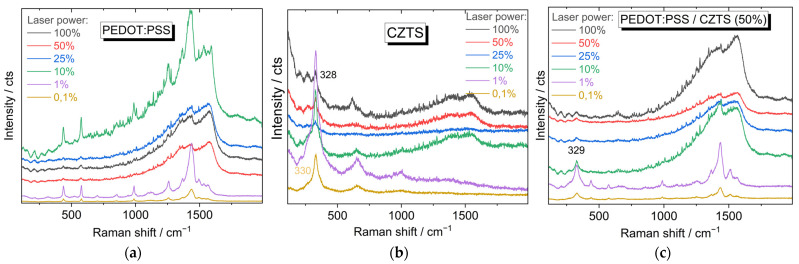
Raman spectra at different laser powers, as recorded for (**a**) PEDOT:PSS; (**b**) CZTS NCs; and (**c**) their composite.

**Table 1 nanomaterials-13-00041-t001:** Results of TE measurements (performed at a temperature of 310 K, when neither polymer nor NC films have deteriorated). The samples were deposited by spin-coating on glass substrates and an FLA power density was chosen to be 3 J/cm2 to avoid the decomposition of the PEDOT:PSS in the composites. The TE properties of the films were assessed via calculation of the power factor P = σS^2^, where σ is the electrical conductivity and S is the Seebeck coefficient.

Sample		Conductivity (σ) [S/cm]	Seebeck Coefficient (S) [µV/K]	Power Factor (P) [µW/cmK^2^]
PEDOT:PSS	Pristine FLA	0.6 2.3	0.44 21.0	0.0000001 0.001
PEDOT:PSS 50% + CZTS 50%	Pristine FLA	1.4 24.0	20.0 37.0	0.0006 0.03
(PEDOT:PSS + DMSO) + 50% CZTS 50%	Pristine FLA	9.0 70.83	29.1 24.14	0.007 0.041

## Data Availability

The data presented in this study are available on request from the corresponding author. The data are not publicly available due to some of the presented results being preliminary results and still in processing and will be the topic of a future paper.

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
