# Peer review of "Raman Spectroscopy and Thermoelectric Characterization of Composite Thin Films of Cu2ZnSnS4 Nanocrystals Embedded in a Conductive Polymer PEDOT:PSS"

_nanomaterials, 2022, doi:10.3390/nano13010041_

Round 1
Reviewer 1 Report
My comments are in attachement.

Author Response
Authors reply: we are grateful to the reviewer for careful reading of the manuscript and comments and corrections made for its improvement. We have proofread the English once again, as suggested, implemented all the suggested minor corrections, including also additional references, and give our point-by-point answers to the rest of the comments. All the changes made in the manuscript are shown in the correction mode in the revised manuscript file (please see the attachment).
Page 5
In order to check this, a control sample was prepared as a TGA solution in water, at a concentration that can be expected in the samples of CZTS NCs.
What is a TGA?
Authors reply: The explanation about TGA was added to the text: “The NC solution used for the preparation of composites with PEDOT:PSS were subject to an usual purification procedure that removes most of the residuals of the synthesis, dissolved in water, such as NaCl and other salts, excess of thioglycolic acid (TGA) ligand molecules (i.e. ligand molecules not bound to the NC surface).“
Page 6
The spectrum of FLA-treated CZTS NCs (Figure 4a) exhibits an upward shift of the main band (334 cm-1) compared to the pristine material (331 cm-1) indicating an improvement in the crystallinity of the compound, an expected consequence of FLA treatment.
What is the difference between 334 cm-1 and 331 cm-1, if the spectral resolution is about 2 cm−1 ?
Authors reply: the spectral resolution limits our ability to resolve closely positioned peaks in the spectrum, but the spectral position of each resolved peak can be determined with much better (at least an order of magnitude) precision than the magnitude of the spectral resolution. In our case the peak position could be determined with an error of only 0.2-0.3 cm-1. Therefore, the peak positions at 334 cm-1 and 331 cm-1 can be distinguished reliably. Besides, we also performed up to 5 measurements in different spots on each sample to make sure the measured values are representative of the sample.
Page 7
A probable explanation of why we observe changes in the electrical properties but do not observe any changes in the Raman spectra after DMSO addition, while upon NC addition we observed changes in both Raman spectra and electrical properties, is that NCs have an influence on the electronic structure of the polymer while DMSO just leads to an ordering of the PE- DOT:PSS molecules.
This long and complex sentence could be divided for clarity.
Authors reply: We split the sentence into three: “Therefore, we observe changes in the electrical properties but do not observe any changes in the Raman spectra after DMSO addition. Upon NC addition, however, we observed changes in both Raman spectra and electrical properties. A probable explanation is that NCs have an influence on the electronic structure of the polymer while DMSO just leads to an ordering of the PEDOT:PSS molecules.“
Page 8
The improvement of all of the TE properties of such a composite is much better when the corresponding FLA-treated films are compared.
This sentence is unclear.
Authors reply: We modified the sentence as follows: “The superior TE properties of the composite over those of bare NCs and polymer are more pronounced for the corresponding FLA-treated films.“

Reviewer 2 Report
The manuscript presents a Raman spectroscopy study of composite films, prepared from aqueous colloidal solution of Cu2ZnSnS4 nanocrystals (CZTS NCs) mixed in different proportions with a PEDOT:PSS solution. Such films are promising for photovoltaic and thermoelectric applications and can help in solving the existing challenge with converting the heat generated by solar cells into electricity. The focus of the manuscript is on the investigation of the structural stability of both components in the composite thin films with different NC-to-polymer ratios. Moreover, stability of the composite films subjected to flash-lamp annealing or laser treatment was analyzed. It was demonstrated that CZTS NCs and PEDOT:PSS mutually influence each other in the composite. The composite films revealed enhanced thermoelectrical properties compared to the pure PEDOT:PSS. Further improvement of thermoelectrical properties was done by flash-lamp annealing and by addition of DMSO.
The manuscript suits the topics of Nanomaterials and can be interesting for the readership of the journal as it presents new results on the fabrication and investigation of thermoelectric nanocomposite films. I recommend the publication of this manuscript in Nanomaterials after the following minor revisions:
1. In “Materials and Methods” section, the authors should describe in more detail the thickness measurement of the specimens by AFM.
2. In line 101, PPS must be changed to PSS.
Author Response
We are grateful to the reviewer for careful reading of the manuscript and comments made for its improvement. We have proofread the English once again, as suggested, while our point-by-point answers to the rest of the comments are given below. All the changes made in the manuscript are shown in the correction mode in the revised manuscript file (please see the attachment).
- In “Materials and Methods” section, the authors should describe in more detail the thickness measurement of the specimens by AFM.
Authors reply: In the section “Materials and Methods” after the sentence “To correctly determine the electrical conductivity, the thickness of each sample was measured using atomic force microscopy (AFM Agilent 5500).“ a sentence was added with a more detailed description of the procedure: “To do this, a scratch was made on the film using a sharp object and a force enough to reach the substrate without damaging it. The AFM profile on the scratch is then measured and the depth (height) of the scratch is considered as the thickness of the film.”
- In line 101, PPS must be changed to PSS.
Authors reply: Corrected.

Reviewer 3 Report
The manuscript entitled " Raman spectroscopy and thermoelectric characterization of composite thin films of Cu2ZnSnS4 nanocrystals embedded in a conductive polymer PEDOT:PSS'' has been investigated in detail. The topic addressed in the manuscript is potentially interesting and the manuscript contains some practical meanings. The paper is interesting for readers and the article brings new insights. Overall, this is a clear, concise, and well-written manuscript. The research is original, understandable, correct, and appropriate for Nanomaterials. The title reflects the content of the article, the experimental results are well discussed.
It can be accepted after some minor revisions:
(1) The use of AFM is mentioned in 2 Materials and Methods, and the results are not shown in this manuscript.
(2) Optical image of CZTS NCs film before and after 12J/cm2 FLA should be added in the text.
(3) Figure 4(b) should be labeled with the broad features of amorphous carbon for easy observation.
(4) The manuscript should explain why the Raman of different regions of PEDOT:PS after FLA are different.
(5) Whether the (PEDOT:PSS + DMSO) + 50% CZTS 50% shown in Table 1 should be (PEDOT:PSS + DMSO) 50% + CZTS 50%.
Author Response
We are grateful to the reviewer for careful reading of the manuscript and comments made for its improvement. We have proofread the English once again, as suggested, while our point-by-point answers to the rest of the comments are given below. All the changes made in the manuscript are shown in the correction mode in the revised manuscript file (please see the attachment).
It can be accepted after some minor revisions:
- The use of AFM is mentioned in 2 Materials and Methods, and the results are not shown in this manuscript.
Authors reply: In this study AFM was used only to determine the film thickness, which is needed for analysis of thermoelectric measurements.
- Optical image of CZTS NCs film before and after 12J/cm2 FLA should be added in the text.
Authors reply: The required images are added in Fig. S3.
- Figure 4(b) should be labeled with the broad features of amorphous carbon for easy observation.
Authors reply: corrected.
- The manuscript should explain why the Raman of different regions of PEDOT:PS after FLA are different.
Authors reply: The following explanation was added at the end of section 3.2: “In such a way we can make the assumption, that the presence of CZTS NCs in PEDOT:PSS strongly affects the behaviour of the polymer during the FLA treatment. The difference in the Raman spectra measured in different regions of the film can be caused by an inhomogeneous distribution of CZTS NCs in the composite.”
- Whether the (PEDOT:PSS + DMSO) + 50% CZTS 50% shown in Table 1 should be (PEDOT:PSS + DMSO) 50% + CZTS 50%.
Authors reply: corrected
